# Comparison between Two Algorithms for Computing the Weighted Generalized Affinity Coefficient in the Case of Interval Data

Áurea Sousa [1,*] , Osvaldo Silva [2] , Leonor Bacelar-Nicolau [3] , João Cabral [4,5] and Helena Bacelar-Nicolau [6]

[1] Faculty of Sciences and Technology, CEEAplA and OSEAN, Universidade dos Açores, 9500-321 Ponta Delgada, Portugal
[2] Faculty of Sciences and Technology, CICSNOVA.UAc, Universidade dos Açores, 9500-321 Ponta Delgada, Portugal; osvaldo.dl.silva@uac.pt
[3] Faculty of Medicine, Institute of Preventive Medicine and Public Health & ISAMB/FM-UL, Universidade de Lisboa, 1649-028 Lisboa, Portugal; lnicolau@medicina.ulisboa.pt
[4] Faculty of Sciences and Technology, Universidade dos Açores, 9500-321 Ponta Delgada, Portugal; joao.mg.cabral@uac.pt
[5] CIMA-Research Centre, Mathematics and Applications & Azores University, 9500-321 Ponta Delgada, Portugal
[6] Faculty of Psychology, Institute of Environmental Health (ISAMB/FM-UL), Universidade de Lisboa, 1649-013 Lisboa, Portugal; hbacelar@psicologia.ulisboa.pt
[*] Correspondence: aurea.st.sousa@uac.pt; Tel.: +351-296-650-073

**Abstract:** From the affinity coefficient between two discrete probability distributions proposed by Matusita, Bacelar-Nicolau introduced the affinity coefficient in a cluster analysis context and extended it to different types of data, including for the case of complex and heterogeneous data within the scope of symbolic data analysis (SDA). In this study, we refer to the most significant partitions obtained using the hierarchical cluster analysis (h.c.a.) of two well-known datasets that were taken from the literature on complex (symbolic) data analysis. h.c.a. is based on the weighted generalized affinity coefficient for the case of interval data and on probabilistic aggregation criteria from a *VL* parametric family. To calculate the values of this coefficient, two alternative algorithms were used and compared. Both algorithms were able to detect clusters of macrodata (aggregated data into groups of interest) that were consistent and consonant with those reported in the literature, but one performed better than the other in some specific cases. Moreover, both approaches allow for the treatment of large microdatabases (non-aggregated data) after their transformation into macrodata from the huge microdata.

**Keywords:** interval data; hierarchical cluster analysis; weighted generalized affinity coefficient; discrete probability distributions



## 1. Introduction

Within the scope of bidimensional data matrices, each variable only takes one single value (for each individual $k$, $Y_j(k)$ can be denoted by $x_{kj}$). However, in today's big data era, "data analysts are confronted with new challenges: they are asked to process data that go beyond the classical framework, as in the case of data concerning more or less homogeneous classes or groups of individuals (second-order objects) instead of single individuals (first-order objects)" ([1] (p. 473)). Moreover, logging large datasets into large databases often leads to the need to summarize these data, considering their underlying concepts, which can only be described by more complex types of data, namely, symbolic data.

In a symbolic data table, the variables can take values as a single quantitative value, a single categorical value, a set of values or categories (multivalued variable), an interval, or a set of values with associated weights. Moreover, the cells may contain data of several

types that can be weighted and linked using logical rules and taxonomies [2]. Furthermore, the statistical units can be either simple elements (e.g., subjects/individuals) or subsets of objects in some population (e.g., subsamples of a sample, classes of a partition, and subgroups of the population). The rows of a symbolic data table correspond to symbolic objects (SOs), a "new type of statistical data that are characterized by their complexity" ([3] (p. 125)). Reviews of several currently available methods for analyzing such data can be found in the existing literature in this field (e.g., [4–14]). Moreover, SOs can be visualized using the Zoom Star Representation (2D Zoom Star and 3D Zoom Star) for graphical representation of multidimensional symbolic data [3,15].

Symbolic data analysis provides a framework for the representation and analysis of data with inherent variability. Therefore, "new variable types have been introduced, whose realizations are not single real values or categories, but sets, intervals, or, more generally, distributions over a given domain" ([8] (p. 282)). In this context, for instance, Billard and Diday [6] looked at the concept of symbolic data in general, including multivalued variables, interval-valued variables, modal multivalued variables, and modal interval-valued (histogram-valued) variables. The present study focuses on interval-valued variables, for which the formal definition is presented below.

Let E = {1, . . ., $N$} be a set of $N$ statistical units (individuals, classes, objects, etc.). A variable $Y_j$ with domain $I_j$ is an interval-valued variable if, for all $k \in E$, $Y_j$ ($k$) is an interval of $I_j$ in the order established on $I_j$. Formally, in this case, Y is mapping E→T (defined on E) so that $I_j$ ($k$) = [$a_{kj}$, $b_{kj}$]. Some examples of interval-valued data or simply interval data are "daily weather temperature, weekly price variations of fish, record of blood pressure of a patient" ([16] (p. 45)), among other examples. Moreover, observations of this type are frequent in cases "such as those involving fluctuations, subjective perceptions, intervals, censored or grouped data" ([17] (p. 229)).

According to Brito et al. [18], interval data occur in various contexts and are often generated from the aggregation of large databases into groups of interest when the individual observations (the microdata) are described using quantitative variables. According to the same authors, when describing ranges of variable values (for instance, for daily stock prices or temperature ranges), we obtain native interval data.

The purpose of cluster analysis is to group either the data units (subjects/persons/cases) or the variables into clusters so that "the elements within a cluster have a high degree of "natural association" among themselves while the clusters are "relatively distinct from one another" ([19] (p. xi)). In the same way, the aim of using clustering methods for symbolic data is to classify the entities into clusters (or classes), "which are internally as homogeneous as possible and externally as distinct from each other as possible" ([6] (p. 482)). According to Brito ([20] (p. 231)), in this context, "the problem consists in developing methods that allow to cluster sets of symbolic data and that produce classes directly interpretable in terms of the input variables".

Several clustering algorithms have been also proposed in the literature. Ezugwu et al. [21] conducted a systematic review of traditional and state-of-the-art clustering techniques for different domains. Moreover, some dissimilarity measures for interval data can be found in the literature concerning symbolic data (e.g., [16,22–28]), and there are also some similarity measures that can deal with interval data (e.g., [17,29–31]). Gordon [32] (p. 141) states that "given a measure of pairwise dissimilarity between symbolic objects, classifications of them can be obtained using standard algorithms that analyse dissimilarity matrices" (the same is valid for a measure of pairwise similarity). Thus, given a proximity matrix, classifications of symbolic objects can be provided applying classical agglomerative algorithms.

In the present paper, we refer to the most significant partitions obtained via applying hierarchical cluster analysis (h.c.a.) on two datasets (interval data) issued from the literature on symbolic data analysis (SDA). h.c.a. is based on the weighted generalized affinity coefficient (e.g., [30]) for the case of interval data and on one classic (single linkage (*SL*)) and two probabilistic aggregation criteria from a *VL* parametric family (e.g., [33–39]). To

calculate the values of this similarity coefficient, two alternative algorithms, which will be explained in Section 2, were used and compared. Section 3 provides a description of the experimental methodology, while Section 4 describes our experimental results, as well as our interpretation of the results and the experimental conclusions that can be drawn from them. Finally, Section 5 (Discussion and Conclusions section) presents our final remarks concerning the two applied algorithms.

## 2. Literature Review: Affinity Coefficient in the Case of Interval Variables

The affinity coefficient was first proposed by Matusita [40,41], who studied its properties and applications mostly in classical statistics. Given two discrete probability distributions, namely, $P = (p_1, \cdots, p_m)$ and $Q = (q_1, \cdots, q_m)$, on $\{1, \cdots, m\}$, the affinity coefficient between them is given by $a(P, Q) = \sum_{v=1}^{m} \sqrt{p_v q_v}$ and relates to a special case of the Hellinger distance, $d(P, Q)$, designated by Bhattacharyya distance via the following formula: $d(P, Q)^2 = 2(1 - a(P, Q))$ (e.g., [42]). From the affinity coefficient between two discrete probability distributions proposed by Matusita [40,41], Bacelar-Nicolau (e.g., [39]) introduced the affinity coefficient as a similarity coefficient between pairs of variables or subjects in a cluster analysis context. Later, this coefficient was extended to different types of data, including complex and heterogeneous data (e.g., [30,42]), within the context of SDA.

Given a pair of two statistical units $(k, k' = 1(1)N)$, the extension of the affinity coefficient for the case of symbolic data, called the weighted generalized affinity coefficient, is given by the following formula, which is presented in [30] (p. 11) (the notation $k, k' = 1(1)N$ means that $k$ and $k'$ vary from 1 to $N$ by integer values):

$$a(k, k') = \sum_{j=1}^{p} \pi_j aff\left(k, k'; j\right) = \sum_{j=1}^{p} \pi_j \sum_{\ell=1}^{m_j} \sqrt{\frac{x_{kj\ell}}{x_{kj\bullet}} \cdot \frac{x_{k'j\ell}}{x_{k'j\bullet}}}, \tag{1}$$

where $aff(k, k'; j))$ denotes the generalized local affinity between the two statistical units, $k$ and $k'$, over the $j$th variable; $m_j$ is the number of columns of a generalized sub-table associated with the $j$th variable; $x_{kj\ell}$ designates a positive or null real number (that is, $x_{kj\ell} \in \mathbb{R}_0^+$), for which its meaning is determined by the type of variable $Y_j$ (a proper adaptation of Formula (1) makes them capable of dealing with negative values); $x_{kj\bullet} = \sum_{\ell=1}^{m_j} x_{kj\ell}$, $x_{k'j\bullet} = \sum_{\ell=1}^{m_j} x_{k'j\ell}$, and $\pi_j$, with $j = 1(1)p$, are weights that satisfy constraints $0 \leq \pi_j \leq 1$ and $\sum_{j=1}^{p} \pi_j = 1$. Both coefficients, $(aff(k, k'; j)$ and $a(k, k'))$, assume values in the interval [0,1]. Moreover, the coefficient given by Formula (1) satisfies properties that indicate that it is a robust similarity measure (e.g., [30,42]). It should be noted that, in particular, when the initial symbolic data matrix contains absolute frequencies (counts), this coefficient deals with discrete data (the counts can be mapped in one-to-one correspondence with the set of positive or null integers), and in this situation, we consider notations $n_{kj\ell}$, $n_{k'j\ell}$, $n_{kj\bullet}$ and $n_{k'j\bullet}$ instead of $x_{kj\ell}$, $x_{k'j\ell}$, $x_{kj\bullet}$, and $x_{k'j\bullet}$, respectively (see [42]). In this particular case, $n_{kj\ell}$ is the number of individuals (in the unit $k$) that share category $\ell$ of variable $Y_j$ and $n_{kj\bullet} = \sum_{\ell=1}^{m_j} n_{kj\ell}$ (similarly, $n_{k'j\bullet} = \sum_{\ell=1}^{m_j} n_{k'j\ell}$). In this context, it is important to emphasize that the relative frequencies $\frac{x_{kj\ell}}{x_{kj\bullet}}$, $\ell = 1(1)m_j$, and $\frac{x_{k'j\ell}}{x_{k'j\bullet}}$, $\ell = 1(1)m_j$ generate two discrete distributions, with the "square root profiles" $\left(\sqrt{p_{kj1}}, \cdots, \sqrt{p_{kjm_j}}\right)$ and $\left(\sqrt{p_{kj'1}}, \cdots, \sqrt{p_{kj'm_j}}\right)$, respectively. In addition, the weighted generalized affinity coefficient, $a(k, k')$, measures the monotone tendency between two square root profiles of a pair of statistical units, $k$ and $k'$.

In the case of a symbolic data table in which the values are intervals, the calculation of the weighted generalized affinity coefficient can be carried out based on the algorithms described below.

### 2.1. Algorithm 1: Computation Directly from the Initial Intervals

Let E = {1, ..., N} be a set of N statistical units described by $p$ interval-valued variables, $Y_j$, with $j = 1(1)p$, which takes values in the interval $I_{kj}$, $(k = 1(1)N)$ of $\mathbb{R}$. Therefore, each element of E is represented by a $p$-dimensional vector of intervals, as shown in Table 1.

**Table 1.** Symbolic data table (interval data).

|   | $Y_1$ | $\cdots$ | $Y_j$ | $\cdots$ | $Y_p$ |
|---|---|---|---|---|---|
| 1 | $I_{11} = [a_{11}, b_{11}]$ | $\cdots$ | $I_{1j} = [a_{1j}, b_{1j}]$ | $\cdots$ | $I_{1p} = [a_{1p}, b_{1p}]$ |
| $\vdots$ | $\vdots$ | $\vdots$ | $\vdots$ | $\vdots$ | $\vdots$ |
| $k$ | $I_{k1} = [a_{kj}, b_{kj}]$ | $\cdots$ | $I_{kj} = [a_{kj}, b_{kj}]$ | $\cdots$ | $I_{kp} = [a_{kp}, b_{kp}]$ |
| $\vdots$ | $\vdots$ | $\vdots$ | $\vdots$ | $\vdots$ | $\vdots$ |
| $N$ | $I_{N1} = [a_{N1}, b_{N1}]$ | $\cdots$ | $I_{Nj} = [a_{Nj}, b_{Nj}]$ | $\cdots$ | $I_{Np} = [a_{Np}, b_{Np}]$ |

The entry $I_{kj} = [a_{kj}, b_{kj}]$ of Table 1, corresponding to the description of the data unit $k$ ($k = 1(1)N$) in the variable $Y_j$ ($j = 1(1)p$), contains an interval $I_{kj}$ of $\mathbb{R}$, for which its lower and upper boundaries are denoted by $a_{kj}$ and $b_{kj}$, respectively. Thus, each entry of this table is defined as a closed and bounded interval, and they are often used to represent a quantity that may vary between an upper boundary and a lower boundary [16]. In this scenario, the local generalized affinity coefficient, $aff\left(k, k'; j\right)$, between a pair of statistical units, $k$ and $k'$ ($k, k' = 1(1)N$), with respect to variable $Y_j$ can be computed directly using Formula (2), which was initially presented in [30] (p. 15), where $\left|I_{kj}\right|$, $\left|I_{k'j}\right|$, and $\left|I_{kj} \cap I_{k'j}\right|$ denote the ranges of intervals $I_{kj}$, $I_{k'j}$, and $I_{kj} \cap I_{k'j}$ (intersection of the intervals).

$$aff\left(k, k'; j\right) = \frac{\left|I_{kj} \cap I_{k'j}\right|}{\sqrt{\left|I_{kj}\right| \cdot \left|I_{k'j}\right|}} \qquad (2)$$

The weighted generalized affinity coefficient between $k$ and $k'$ ($k, k' = 1(1)N$) can be computed using Formula (3):

$$a\left(k, k'\right) = \sum_{j=1}^{p} \pi_j \cdot aff(k, k'; j) = \sum_{j=1}^{p} \pi_j \cdot \frac{\left|I_{kj} \cap I_{k'j}\right|}{\sqrt{\left|I_{kj}\right| \cdot \left|I_{k'j}\right|}}, \qquad (3)$$

where the weights, $\pi_j$, satisfy two conditions, namely, $0 \leq \pi_j \leq 1$ and $\sum_{j=1}^{p} \pi_j = 1$. In Algorithm 1, we directly use Formula (3) without any decomposition of the initial intervals (e.g., [30]). In other words, the input of algorithm 1 is the data matrix corresponding to Table 1 (more specifically, the lower and upper boundaries of the respective intervals, the number of statistical units ($N$), and the number of variables).

Given two intervals, A = [*LowerA, UpperA*] and B = [*LowerB, UpperB*], where *Lower A* and *UpperA* denote the lower and upper boundaries of A (analogous notation for B), respectively, it is important to highlight that the computation of the local generalized affinity coefficient, $aff(k, k'; j)$, between a pair of statistical units, $k$ and $k'$ ($k, k' = 1(1)N$), concerning variable $Y_j$, according to Formula (2), can be implemented by considering the following three steps: (i) the computation of the ranges of A and B; (ii) the intersection between A and B (here, *Inters1* and *Inters2* are the lower and upper boundaries of the interval corresponding to the intersection of A and B, respectively); and (iii) the computation of $aff\left(k, k'; j\right)$ (abbreviated as *aff*).

---

**Algorithm 1:** Computation Directly from the Initial Intervals

---

*RangeA = UpperA-LowerA*
*RangeB = UpperB-LowerB*
***If** ((UpperA < LowerB). OR. (UpperB < LowerA))* **Then**
*aff = 0 (There is no intersection between intervals A and B)*
***else***
*Inters1 = max(LowerA, LowerB)*
*Inters2 = min(UpperA, UpperB)*
*RangeInt = Inters2-Inters1*
*aff = RangeInt/SQRT(RangeA\*RangeB)*
***End if***

---

*2.2. Algorithm 2: Previous Decomposition of the Initial Intervals into a Set of $m_j$ Elementary and Disjoint Subintervals and the Generation of a New Data Matrix*

Let $I_j$ be the union of the initial intervals, $I_{kj}$ ($k = 1(1)N$), which refers to the description of $N$ statistical units (often groups of individuals) in variable $Yj$ (see the $j$-th column of Table 1); that is, $I_j = \bigcup_{k=1}^{N} I_{kj} = \bigcup_{k=1}^{N} \left[a_{kj}, a_{kj}\right]$ ($k = 1, \ldots, N$) is the domain, $I_j$, of $Y_j$.

The second algorithm consists of the calculation of the general formula of the weighted generalized affinity coefficient (1), considering an appropriate decomposition of the initial intervals, $I_{kj}$ ($k = 1(1)N$), into a set of $m_j$ elementary and disjoint subintervals and working with the ranges of these new intervals. In this approach, in the first step, the domain, $I_j$, of each variable $Y_j$ ($j = 1(1)N$) is decomposed into a set of $m_j$ elementary and disjoint intervals $\left\{I_{j\ell}; \ell = 1, \cdots, m_j\right\}$ so that $\ell, \ell' = 1(1)m_j, \ell \neq \ell'; k = 1(1)N : I_j = \bigcup_{\ell=1}^{m_j} I_{j\ell}$; $\left|I_{j\ell} \cap I_{j\ell'}\right| = 0$; and

$$\left|I_{kj} \cap I_{j\ell}\right| = \begin{cases} \left|I_{j\ell}\right|, \text{ if } \left|I_{kj} \cap I_{J\ell}\right| \neq 0 \\ 0, \text{ otherwise} \end{cases},$$

where | | symbolizes the interval range (difference between the upper and the lower boundaries of the corresponding interval).

In the second step, a new data matrix, subdivided into $p$ subtables/submatrices (one for each variable, $Y_j$ ($j = 1(1)p$), is obtained. It should be noted that, in each of these subtables, the $k$-th row corresponds to the description of statistical unit $k$ in terms of the ranges of the intersection between the initial interval $I_{kj}$ ($k = 1(1)N$) and each of the $m_j$ elementary subintervals used in the decomposition of domain $I_j$ of variable $Y_j$ ($j = 1(1)p$). Thus, the vector $[[a_{1j}, b_{1j}], \cdots, [a_{kj}, b_{kj}], \cdots, [a_{Nj}, b_{Nj}]]^T$, corresponding to $Y_j$ in Table 1, is replaced by an appropriate subtable, as shown in Table 2.

**Table 2.** Part of the transformed data matrix concerning variable $Yj$ ($j = 1(1)p$)—Algorithm 2.

| | $\cdots$ | | | $Y_j$ | | | $\cdots$ |
|---|---|---|---|---|---|---|---|
| | $\cdots$ | $I_{j1}$ | $\cdots$ | $I_{j\ell}$ | $\cdots$ | $I_{jm_j}$ | $\cdots$ |
| 1 | $\cdots$ | $\left|I_{1j} \cap I_{j1}\right|$ | $\cdots$ | $\left|I_{1j} \cap I_{j\ell}\right|$ | $\cdots$ | $\left|I_{1j} \cap I_{m_j}\right|$ | $\cdots$ |
| $\vdots$ | $\vdots$ | $\vdots$ | $\vdots$ | $\vdots$ | $\vdots$ | $\vdots$ | $\vdots$ |
| $k$ | $\cdots$ | $\left|I_{kj} \cap I_{j1}\right|$ | $\cdots$ | $\left|I_{kj} \cap I_{j\ell}\right|$ | $\cdots$ | $\left|I_{kj} \cap I_{m_j}\right|$ | $\cdots$ |
| $\vdots$ | $\vdots$ | $\vdots$ | $\vdots$ | $\vdots$ | $\vdots$ | $\vdots$ | $\vdots$ |
| $N$ | $\cdots$ | $\left|I_{Nj} \cap I_{j1}\right|$ | $\cdots$ | $\left|I_{Nj} \cap I_{j\ell}\right|$ | $\cdots$ | $\left|I_{Nj} \cap I_{m_j}\right|$ | $\cdots$ |

Finally, in the third step, Formula (1) is applied (for details, see [30]).

When we are dealing with interval-valued variables, Formula (3) arises as a particular case of Formula (1) when considering $m_j$ as equal to the number of elementary and disjoint subintervals of variable $Y_j$ (instead of the number of modalities of the sub-table associated

with the $j$th variable), and the values of $x_{kj\ell}$ and $x_{k'j\ell}$ ($\ell = 1(1)m_j$) are, in this case, the ranges of the corresponding elementary and disjoint subintervals, respectively (note that, in this context, these ranges are a set of countable or enumerable values). Thus, in the second algorithm, we can formally define $x_{kj\ell}$ and $x_{k'j\ell}$ as follows:

$$x_{kj\ell} = \begin{cases} \left| I_{j\ell} \right|, & \text{if } I_{kj} \cap I_{j\ell} = I_{j\ell} \\ 0, & \text{otherwise} \end{cases}$$

and

$$x_{k'j\ell} = \begin{cases} \left| I_{j\ell} \right|, & \text{if } I_{k'j} \cap I_{j\ell} = I_{j\ell} \\ 0, & \text{otherwise} \end{cases}$$

It should also be noted that both in Formula (3) and Formula (1), $x_{kj\bullet} = \sum_{\ell=1}^{m_j} x_{kj\ell}$, $x_{k'j\bullet} = \sum_{\ell=1}^{m_j} x_{k'j\ell}$, and $\pi_j$, with $j = 1(1)p$, are weights such as $0 \leq \pi_j \leq 1$ and $\sum_{j=1}^{p} \pi_j = 1$. Moreover, we have the following: $x_{kj\bullet} = \sum_{\ell=1}^{m_j} x_{kj\ell} = \left| I_{kj} \right|$; $x_{k'j\bullet} = \sum_{\ell=1}^{m_j} x_{k'j\ell} = \left| I_{k'j} \right|$, and $\sum_{\ell=1}^{m_j} x_{kj\ell} x_{k'j\ell} = \left| I_{kj} \cap I_{k'j} \right|$. Thus, if some conditions are verified (namely, (i) if there are no intervals with identical lower and upper boundaries and (ii) if there are no intervals with an intersection that is a single point), the application of Formula (1), considering a data matrix as exemplified in Table 2, provides the same affinity values as Algorithm 1 (Formula (3)). In those conditions, we have the following (for details, see, [30]):

$$a\left(k, k'\right) = \sum_{j=1}^{p} \pi_j aff\left(k, k'; j\right) = \sum_{j=1}^{p} \pi_j \sum_{\ell=1}^{m_j} \sqrt{\frac{x_{kj\ell}}{x_{kj\bullet}} \cdot \frac{x_{k'j\ell\cdot}}{x_{k'j\bullet}}} = \sum_{j=1}^{p} \pi_j \cdot \frac{\left| I_{kj} \cap I_{k'j} \right|}{\sqrt{\left| I_{kj} \right| \cdot \left| I_{k'j} \right|}}$$

When there are initial intervals with identical lower and upper boundaries, in both described algorithms, we can obtain transformed intervals by replacing "these intervals by transformed intervals obtained from the first ones, for instance, by subtracting and adding 0.5, respectively to the lower and upper boundaries" (this procedure is well illustrated in [43] (p. 17)).

When the variables assume values of different magnitudes and scales, we recommend the use of the asymptotically centered and reduced coefficient under a permutational hypothesis of reference based on the Wald and Wolfowitz limit theorem, as denoted by $a_{WW}(k, k')$ (e.g., [30]). The coefficient $a_{WW}(k, k')$, in turn, allows for the definition of a probabilistic coefficient, $\alpha_{WW}(k, k')$, in the context of the *VL* methodology (*V* for validity and *L* for linkage) along the lines initiated by Lerman ([33–37]) and developed by Bacelar-Nicolau and Nicolau (e.g., [38,39]). The application of $\alpha_{WW}(k, k')$ instead of $a(k, k')$ allows us to deal with comparable similarity values using a probabilistic scale. Several applications of this methodology using well-known datasets concerning interval data in the context of SDA can be found in the literature (for example, in [30,43–45]).

## 3. Materials and Methods

The main objective of the empirical part of the present study was to understand and illustrate the situations in which one algorithm should be used over the other. For this purpose, we used two datasets from the literature on symbolic data (the Abalone dataset and City temperature interval dataset), both of which are described below.

The Abalone dataset concerns 4177 cases of marine crustaceans which are described according to nine attributes (e.g., [1,45]): sex; length (longest shell measurement) in mm; diameter (perpendicular to length) in mm; height (measured with meat in shell) in mm; whole weight (weight of the whole abalone) in grams; shucked weight (weight of the meat) in grams; viscera weight (gut weight after bleeding) in grams; shell weight (weight of the dried shell) in grams; and rings (number of rings). The microdata concerning this dataset are available at http://archive.ics.uci.edu/dataset/1/abalone (accessed on 1 May 2023).

In the present study, initially, using the DB2S0 facility available in the SODAS (Symbolic Official Data Analysis System) software [46], version 2.50, nine Boolean symbolic objects (BSOs) were generated. Each of these BSOs correspond to an interval of values for the number of rings of the crustaceans: A (1–3 rings), B (4–6), C (7–9), D (10–12), E (13–15), F (16–18), G (19–21), H (22–24), and I (25–29 rings). Abalone data were thus aggregated into BSOs, each of which correspond to a range of values for the number of rings. For each of the groups, we considered seven interval-valued variables, namely, "Length", "Diameter", "Height", "Whole", "Shucked", "Viscera", and "Shell". The symbolic matrix presented in Appendix A (see Table A1) contains a description of the nine groups of abalones using interval-valued variables considering five decimal places (in [1] (p. 478), the entries of this symbolic data matrix are shown with only two decimal places).

According to Malerba et al. [1], it is expected that abalones with the same number of rings should also present similar values for these attributes. Therefore, it is expected that "the degree of dissimilarity between crustaceans computed on the independent attributes to actually be proportional to the dissimilarity in the dependent attribute (i.e., difference in the number of rings)" ([1] (p. 477)). This property is known as "Monotonic Increasing Dissimilarity" (abbreviated as MID property). Moreover, concerning the prediction tasks, the number of rings is the value to be predicted from which it is possible to know the age in years of the crustacean by adding 1.5 to the number of rings. Thus, this dataset is characterized by a fully understandable and explainable property (the MID property).

The second dataset analyzed in the present paper concerns the minimum and maximum temperatures in degrees centigrade, which are recorded in 37 cities during a year, as shown in Table A2 of Appendix A. The intuitive partition carried out by a group of human observers resulted in four clusters of cities (e.g., [47]): {2, 3, 4, 5, 6, 8, 11, 12, 15, 17, 19, 22, 23, 29, 31}; {0, 1, 7, 9, 10, 13, 14, 16, 20, 21, 24, 25, 26, 27, 28, 30, 33, 34, 35, 36}; {18}; {32}.

Both algorithms were applied considering $\pi_{jj'} = 1/p$ if $j = j'$ and $\pi_{jj'} = 0$ if $j \neq j'$ in the corresponding mathematical formulas ((3) and (1), respectively). The values of the corresponding similarity matrices were combined with three aggregation criteria, namely, the classic single linkage (*SL*) and two probabilistic (*AV1* and *AVB* (see [33–39])). In the case of the first dataset, the selection of the best partitions was based on the values of Global Statistics of Levels (*STAT*) and *DIF* indexes (e.g., [39]). However, in the case of the second dataset, we directly compared the obtained partitions with the a priori partition.

## 4. Results

### 4.1. Application to Abalone Data

Here, we only applied Algorithm 1 (Formula (3)), as the data matrix does not contain intervals with an intersection that act as a single point, and the variables assume values with magnitudes and scales that do not differ substantially. The values of the weighted generalized affinity coefficient are shown in Table 3.

**Table 3.** Lower triangular similarity matrix (Abalone data): weighted generalized affinity coefficient—Algorithm 1.

|   | A | B | C | D | E | F | G | H | I |
|---|---|---|---|---|---|---|---|---|---|
| A | 1.000000 | | | | | | | | |
| B | 0.295899 | 1.000000 | | | | | | | |
| C | 0.096265 | 0.714581 | 1.000000 | | | | | | |
| D | 0.004515 | 0.611748 | 0.835651 | 1.000000 | | | | | |
| E | 0.000000 | 0.625292 | 0.804286 | 0.884309 | 1.000000 | | | | |
| F | 0.000000 | 0.550538 | 0.735457 | 0.799001 | 0.874737 | 1.000000 | | | |
| G | 0.000000 | 0.541550 | 0.705257 | 0.731724 | 0.815422 | 0.889002 | 1.000000 | | |
| H | 0.000000 | 0.436690 | 0.665697 | 0.748095 | 0.817462 | 0.831564 | 0.871040 | 1.000000 | |
| I | 0.000000 | 0.246183 | 0.498152 | 0.531911 | 0.595100 | 0.660145 | 0.727961 | 0.700374 | 1.000000 |

Figure 1 corresponds to the dendrogram concerning the combination of the weighted generalized affinity coefficient (computed using algorithm 1) with the *AV1* and *AVB* methods.

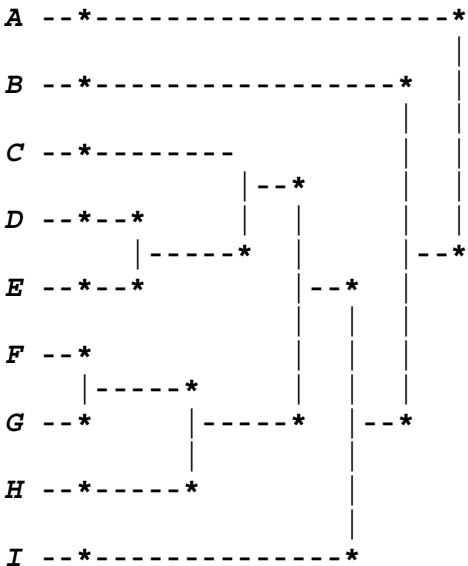

**Figure 1.** Dendrogram provided using *AV1* and *AVB* methods. A (1–3 rings), B (4–6), C (7–9), D (10–12), E (13–15), F (16–18), G (19–21), H (22–24), and I (25–29 rings), respectively.

According to the values of the *STAT* and *DIF* indexes, provided that the *AV1* and *AVB* methods are being used, the best partition contains four clusters (cutoff at level 5), namely, {A}; {B}; {F, G, E, H, D, C}; and {I}, where the first and second clusters, {A} and {B}, are concerned with the younger crustaceans, and the fourth cluster, {I}, is concerned with older crustaceans (see Figure 1). Thus, these results satisfy the MID property. Moreover, the most atypical cluster is cluster {A}, representing very young abalones, which is in line with what is referred to in [1] (p. 480). Furthermore, based on the empirical evaluation of a list of dissimilarity measures proposed in [1] for a restricted class of symbolic data, namely, Boolean symbolic objects, the authors state the following: "only three dissimilarity measures proposed by de Carvalho, namely, SO_1, SO_2 and C_1, satisfy the MID property" (see [1] (p. 480)). In this context, it should be noted that our clustering results are in accordance with the results of these three measures.

### 4.2. Application to City Temperature Interval Dataset

Starting from the principle that cluster analysis is sensitive to differences concerning the scales and magnitudes among the variables, here, we present the main partitions concerning the AHCA of the 37 cities using the probabilistic coefficient, $a_{WW}(k, k')$, associated with the asymptotic standardized weighted generalized affinity coefficient, which is obtained using the method of Wald and Wolfowitz. In this context, the decomposition of the initial data matrix in a new submatrix is presented in [44] for the case of variable $Y_1$ (January).

The partition into four clusters provided using the *SL* method (see Figure 2) is identical to that provided by the panel of human observers (a priori partition), which was also obtained by Guru et al. ([47]).

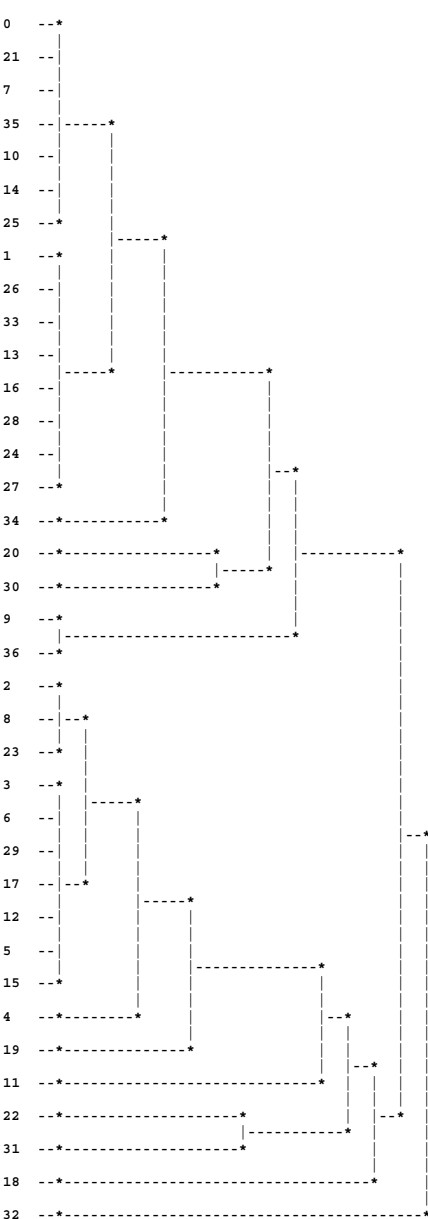

**Figure 2.** Last levels of the dendrogram provided using *SL* and $\alpha_{WW}(k, k')$ methods. The numbers 0 to 36 correspond to the 37 cities.

The partition into four clusters provided by the remaining used aggregation criteria did not coincide with the a priori partition, but this fact is not surprising because other authors (e.g., [48]), through using other algorithms, have also reported partitions of four clusters that were not identical to the a priori partition given by the panel of human observers. However, the following three cluster partitions (provided using the $a_{WW}(k, k')$ coefficient combined with all applied aggregation criteria) only differs from the a priori partition in terms of the inclusion of city 18 (according to the panel of human observers, this city is a singleton):

Cluster 1: {2, 3, 4, 5, 6, 8, 11, 12, 15, 17, 18, 19, 22, 23, 29, 31};
Cluster 2: {0, 1, 7, 9, 10, 13, 14, 16, 20, 21, 24, 25, 26, 27, 28, 30, 33, 34, 35, 36};
Cluster 3: {32}.

## 5. Discussion and Conclusions

In this paper, we addressed the problem of clustering interval data by applying two algorithms for the computation of the weighted generalized affinity coefficient.

The first algorithm works directly with the initial intervals by applying Formula (3), while the second one implies the previous calculation of a new data matrix based on the decomposition of the initial intervals into several elementary and disjoint subintervals followed by the application of Formula (1). Therefore, Algorithm 1 requires less computational effort. However, the second algorithm performs better when there are intervals with an intersection that act as a single point or when the variables assume values of different magnitudes and scales (in the case of the City temperature interval dataset; see the previous section). In this context, we usually opt for using the asymptotically centered and reduced coefficient under a permutational hypothesis of reference that is based on the Wald and Wolfowitz limit theorem, which is denoted by $a_{WW}(k, k')$, or by the corresponding probabilistic coefficient, $\alpha_{WW}(k, k')$, in the context of the *VL* methodology. This last similarity coefficient has the advantage of dealing with comparable similarity values on a probabilistic scale. Furthermore, in the remaining situations, the two algorithms provide the same values for the weighted generalized affinity coefficient.

In the first analyzed dataset (Abalone data), interval data resulted from the aggregation of microdata, while in the second dataset (City temperature interval dataset), the data matrix was already obtained using condensed information in the form of intervals. Furthermore, when there are some intervals with identical lower and upper boundaries, these intervals may be replaced by suitable transformed intervals in both algorithms (1 and 2), as we have mentioned previously. Moreover, both approaches allow for the treatment of large microdata bases (non-aggregated data) by previously generating macrodata (aggregated data into the groups of interest) from the huge microdata. Finally, both algorithms were able to detect the clusters of macrodata that were consistent and concordant with those reported in the literature.

**Author Contributions:** Conceptualization, Á.S., H.B.-N. and O.S.; data curation, Á.S. and O.S.; formal analysis, Á.S. and H.B.-N.; funding acquisition, Á.S. and J.C.; investigation, Á.S., H.B.-N., O.S. and L.B.-N.; methodology, Á.S. and H.B.-N.; software, Á.S. and O.S.; supervision, Á.S and H.B.-N.; writing—original draft, Á.S. and O.S.; Writing—review and editing, H.B.-N., L.B.-N. and J.C. All authors have read and agreed to the published version of the manuscript.

**Funding:** This research article was financed by Portuguese national funds through FCT—Fundação para a Ciência e a Tecnologia, I.P. (project number UIDB/00685/2020).

**Institutional Review Board Statement:** Not applicable.

**Informed Consent Statement:** Not applicable.

**Data Availability Statement:** Not applicable.

**Conflicts of Interest:** The authors declare no conflict of interest.

**Appendix A**

**Table A1.** Abalone data * (four decimal places).

| Length | Diameter | Height | Whole | Shucked | Viscera | Shell |
|---|---|---|---|---|---|---|
| [0.0750, 0.2400] | [0.0550, 0.1750] | [0.0100, 0.0650] | [0.0020, 0.0665] | [0.0010, 0.0310] | [0.0005, 0.0135] | [0.0015, 0.0170} |
| [0.1300, 0.6600] | [0.0950, 0.4750) | [0.0000, 0.1800] | [0.0105, 1.3695] | [0.0050, 0.6410] | [0.0005, 0.2940] | [0.0035, 0.3505] |
| [0.2050, 0.7450] | [0.1550, 0.5800] | [0.0000, 1.1300] | [0.0425, 2.3305] | [0.0170, 1.2530] | [0.0055, 0.5410] | [0.0155, 0.5580] |
| [0.2900, 0.7800 | [0.2250, 0.6300] | [0.0600, 0.5150] | [0.1200, 2.7795] | [0.0415, 1.4880] | [0.0240, 0.7600] | [0.0400, 0.7260] |
| [0.3200, 0.8150] | [0.2450, 0.6500] | [0.0800, 0.2500] | [0.1585, 2.5500] | [0.0635,1.3510] | [0.0325, 0.5750] | [0.0500, 0.7975] |
| [0.3950, 0.7750] | [0.3150, 0.6000] | [0.1050, 0.2400] | [0.3515, 2.8255] | [0.1135, 1.1465] | [0.0565, 0.4805] | [0.1195, 1.005] |
| [0.4500,0.7350] | [0.3550, 0.5900] | [0.1200, 0.2300] | [0.4120, 2.1300] | [0.1145, 0.8665] | [0.0665, 0.4900] | [0.1600, 0.8500] |
| [0.4500, 0.8000] | [0.3800, 0.6300] | [0.1350, 0.2250] | [0.6400, 2.5260] | [0.1580, 0.9330] | [0.1100, 0.5900] | [0.2400, 0.7100] |
| [0.5500, 0.7000] | [0.4650, 0.5850] | [0.1800, 0.2250] | [1.0575, 2.1835] | [0.3245, 0.7535] | [0.1900, 0.3910] | [0.3750, 0.8850] |

* The nine rows corresponding to the abalones A (1–3 rings), B (4–6), C (7–9), D (10–12), E (13–15), F (16–18), G (19–21), H (22–24), and I (25–29 rings), respectively.

**Table A2.** City temperature interval dataset.

| | Cities | Jan. | Feb. | Mar. | Apr. | May | Jun. | Jul. | Aug. | Sept. | Oct. | Nov. | Dec. |
|---|---|---|---|---|---|---|---|---|---|---|---|---|---|
| 0 | Amsterdam | [−4, 4] | [−5, 3] | [2, 12] | [5, 15] | [7, 17] | [10, 20] | [10, 20] | [12, 23] | [10, 20] | [5, 15] | [1, 10] | [−1, 4] |
| 1 | Athens | [6, 12] | [6, 12] | [8, 16] | [11, 19] | [16, 25] | [19, 29] | [22, 32] | [22, 32] | [19, 28] | [16, 23] | [11, 18] | [8, 14] |
| 2 | Bahrain | [13, 19] | [14, 19] | [17, 23] | [21, 27] | [25, 32] | [28, 34] | [29, 36] | [30, 36] | [28, 34] | [24, 31] | [20, 26] | [15, 21] |
| 3 | Bombay | [19, 28] | [19, 28] | [22, 30] | [24, 32] | [27, 33] | [26, 32] | [25, 30] | [25, 30] | [24, 30] | [24, 32] | [23, 32] | [20, 30] |
| 4 | Cairo | [8, 20] | [9, 22] | [11, 25] | [14, 29] | [17, 33] | [20, 35] | [22, 36] | [22, 35] | [20, 33] | [18, 31] | [14, 26] | [10, 20] |
| 5 | Calcutta | [13, 27] | [16, 29] | [21, 34] | [24, 36] | [26, 36] | [26, 33] | [26, 32] | [26, 32] | [26, 32] | [24, 32] | [18, 29] | [13, 26] |
| 6 | Colombo | [22, 30] | [22, 30] | [23, 31] | [24, 31] | [25, 31] | [25, 30] | [25, 29] | [25, 29] | [25, 30] | [24, 29] | [23, 29] | [22, 30] |
| 7 | Copenhagen | [−2, 2] | [−3, 2] | [−1, 5] | [3, 10] | [8, 16] | [11, 20] | [14, 22] | [14, 21] | [11, 18] | [7, 12] | [3, 7] | [1, 4] |
| 8 | Dubai | [13, 23] | [14, 24] | [17, 28] | [19, 31] | [22, 34] | [25, 36] | [28, 39] | [28, 39] | [25, 37] | [21, 34] | [17, 30] | [14, 26] |
| 9 | Frankfurt | [−10, 9] | [−8, 10] | [−4, 17] | [0, 24] | [3, 27] | [7, 30] | [8, 32] | [8, 31] | [5, 27] | [0, 22] | [−3, 14] | [−8, 10] |
| 10 | Geneva | [−3, 5] | [−6, 6] | [3, 9] | [7, 13] | [10, 17] | [15, 17] | [16, 24] | [16, 23] | [11, 19] | [6, 13] | [3, 8] | [−2, 6] |
| 11 | Hong Kong | [13, 17] | [12, 16] | [15, 19] | [19, 23] | [22, 27] | [25, 29] | [25, 30] | [25, 30] | [25, 29] | [22, 27] | [18, 23] | [14, 19] |
| 12 | Kula Lumpur | [22, 31] | [23, 32] | [23, 33] | [23, 33] | [23, 32] | [23, 32] | [23, 31] | [23, 32] | [23, 32] | [23, 31] | [23, 31] | [23, 31] |
| 13 | Lisbon | [8, 13] | [8, 14] | [9, 16] | [11, 18] | [13, 21] | [16, 24] | [17, 26] | [18, 27] | [17, 24] | [14, 21] | [11, 17] | [8, 14] |
| 14 | London | [2, 6] | [2, 7] | [3, 10] | [5, 13] | [8, 17] | [11, 20] | [13, 22] | [13, 21] | [11, 19] | [8, 14] | [5, 10] | [3, 7] |
| 15 | Madras | [20, 30] | [20, 31] | [22, 33] | [26, 35] | [28, 39] | [27, 38] | [26, 36] | [26, 35] | [25, 34] | [24, 32] | [22, 30] | [21, 29] |
| 16 | Madrid | [1, 9] | [1, 12] | [3, 16] | [6, 19] | [9, 24] | [13, 29] | [16, 34] | [16, 33] | [13, 28] | [8, 20] | [4, 14] | [1, 9] |
| 17 | Manila | [21, 27] | [22, 27] | [24, 29] | [24, 31] | [25, 31] | [25, 31] | [23, 29] | [24, 28] | [25, 28] | [24, 29] | [22, 28] | [22, 27] |
| 18 | Mauritius | [22, 28] | [22, 29] | [22, 29] | [21, 28] | [19, 25] | [18, 24] | [17, 23] | [17, 23] | [17, 24] | [18, 25] | [19, 27] | [21, 28] |
| 19 | Mexico City | [6, 22] | [15, 23] | [17, 25] | [18, 27] | [18, 27] | [18, 27] | [18, 27] | [18, 26] | [18, 26] | [16, 25] | [14, 25] | [8, 23] |
| 20 | Moscow | [−13, −6] | [−12, −5] | [−8, 0] | [0, 8] | [7, 18] | [11, 23] | [13, 24] | [11, 22] | [6, 16] | [1, 8] | [−5, 0] | [−11, −5] |
| 21 | Munich | [−6, 1] | [−5, 3] | [−2, 9] | [3, 14] | [7, 18] | [10, 21] | [12, 23] | [11, 23] | [8, 20] | [4, 13] | [0, 7] | [−4, 2] |
| 22 | Nairobi | [12, 25] | [13, 26] | [14, 25] | [14, 24] | [13, 22] | [12, 21] | [11, 21] | [11, 21] | [11, 24] | [13, 24] | [13, 23] | [13, 23] |
| 23 | New Delhi | [6, 21] | [10, 24] | [14, 29] | [20, 36] | [26, 40] | [28, 39] | [27, 35] | [26, 34] | [24, 34] | [18, 34] | [11, 28] | [7, 23] |
| 24 | New York | [−2, 4] | [−3, 4] | [1, 9] | [6, 15] | [12, 22] | [17, 27] | [21, 29] | [20, 28] | [16, 24] | [11, 19] | [5, 12] | [−2, 6] |
| 25 | Paris | [1, 7] | [1, 7] | [2, 12] | [5, 16] | [8, 19] | [12, 22] | [14, 24] | [13, 24] | [11, 21] | [7, 16] | [4, 10] | [1, 6] |
| 26 | Rome | [4, 11] | [5, 13] | [7, 16] | [10, 19] | [13, 23] | [17, 28] | [20, 31] | [20, 31] | [17, 27] | [13, 21] | [9, 16] | [5, 12] |
| 27 | San Francisco | [6, 13] | [6, 14] | [7, 17] | [8, 18] | [10, 19] | [11, 21] | [12, 22] | [12, 22] | [12, 23] | [11, 22] | [8, 18] | [6, 14] |
| 28 | Seoul | [0, 7] | [1, 6] | [1, 8] | [6, 16] | [12, 22] | [16, 25] | [18, 31] | [16, 30] | [9, 28] | [3, 24] | [7, 19] | [1, 8] |
| 29 | Singapore | [23, 30] | [23, 30] | [24, 31] | [24, 31] | [24, 30] | [25, 30] | [25, 30] | [25, 30] | [24, 30] | [24, 30] | [24, 30] | [23, 30] |
| 30 | Stockholm | [−9, −5] | [−9, −6] | [−4, 2] | [1, 8] | [6, 15] | [11, 19] | [14, 22] | [13, 20] | [9, 15] | [5, 9] | [1, 4] | [−2, 2] |
| 31 | Sydney | [20, 30] | [20, 30] | [18, 26] | [16, 23] | [12, 20] | [5, 17] | [8, 16] | [9, 17] | [11, 20] | [13, 22] | [16, 26] | [20, 30] |
| 32 | Tehran | [0, 5] | [5, 8] | [10, 15] | [15, 18] | [20, 25] | [28, 30] | [36, 38] | [38, 40] | [29, 30] | [18, 20] | [9, 12] | [−5, 0] |
| 33 | Tokyo | [0, 9] | [0, 10] | [3, 13] | [9, 18] | [14, 23] | [18, 25] | [22, 29] | [23, 31] | [20, 27] | [13, 21] | [8, 16] | [2, 12] |
| 34 | Toronto | [−8, −1] | [−8, −1] | [−4, 4] | [−2, 11] | [−8, 18] | [13, 24] | [16, 27] | [16, 26] | [12, 22] | [6, 14] | [−1, 17] | [−5, 1] |
| 35 | Vienna | [−2, 1] | [−1, 3] | [1, 8] | [5, 14] | [10, 19] | [13, 22] | [15, 24] | [14, 23] | [11, 19] | [7, 13] | [2, 7] | [1, 3] |
| 36 | Zurich | [−11, 9] | [−8, 15] | [−7, 18] | [−1, 21] | [2, 27] | [6, 30] | [10, 31] | [8, 25] | [5, 23] | [3, 22] | [0, 19] | [−11, 8] |

Source: "Reprinted/adapted with permission from Guru et al. (2004, p. 1210)" ([47]).

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
