# Peer review of "Comparison between Two Algorithms for Computing the Weighted Generalized Affinity Coefficient in the Case of Interval Data"

_stats, doi:10.3390/stats6040068_

Round 1

Reviewer 1 Report

Report on the manuscript: stats-2579453
Comparison between two Algorithms for Computing the Weighted Generalized Affinity Coefficient in the Case of Interval Data

    From the literature on complex (symbolic) data analysis, the authors of this study highlighted the most important partitions discovered by hierarchical cluster analysis (h.c.a.) of two well-known data sets. For the case of interval data, the h.c.a. is based on the weighted generalized affinity coefficient and on probabilistic aggregation requirements from a VL parametric family. To determine the values of this coefficient, two alternative algorithms were employed and contrasted. Both methods were able to identify macro-data clusters (aggregated data into groups of interest) that were consistent and in line with those published in the literature, but one technique occasionally outperformed the other. Furthermore, both methods allow for the processing of sizable microdata bases (non-aggregated data) following their conversion from sizable microdata to macrodata. The manuscript's content is intriguing, however, I have several significant formal concerns that must be taken into consideration. They are as follows:

1. Despite the beauty of the introduction: Except for three references [16], [17], and [31], all of the references are extremely dated, which is both intriguing and unsettling. a pressing necessity that necessitates revisiting and continuing the topic and its prior investigations, at least for the last five years.
2. The origin of Equations (1) and (3) needs to be supported by a reference, which must be added.
3. I think it's best to put both algorithms into explicit steps that enable the reader to visualize the code. Is this accessible to the authors?
4. Tables 1 and 2 need further clarification.
5. In addition to the fantastic conclusion and discussion section, I think it would be better to include the most notable results as bright spots at the end of the section.

Author Response

Reviewer 1

  1. Despite the beauty of the introduction: Except for three references [16], [17], and [31], all of the references are extremely dated, which is both intriguing and unsettling. a pressing necessity that necessitates revisiting and continuing the topic and its prior investigations, at least for the last Öve years.

We added four references, namely [8, 9, 12, 13]:

Goswami, J.P.; Mahanta, A.K. Interval Data Clustering. IOSR Journal of Computer Engineering (IOSR-JCE) 2020, 22(6), Ser. 1, pp. 45-50. doi: 10.9790/0661-2206014550

Ramos-Guajardo, A.B. A hierarchical clustering method for random intervals based on a similarity measure. Comput Stat 2022, 37, 229-261.

Ezugwu, A.E.; Ikotun, A.M.; Oyelade, O.O..; Abualigah, L.; Agushaka , J.O.; Eke, C.I., Akinyelu, A. S. A comprehensive survey of clustering algorithms: State-of-the-art machine learning applications, taxonomy, challenges, and future research prospects. Engineering Applications of Artificial Intelligence 2022, 110, 104743. https://doi.org/10.1016/j.engappai.2022.10474

Xu, X.; Lu, J..; Wang, W. Hierarchical clustering of complex symbolic data and application for emitter. J. Comput. Sci. TechnoL. 2018, 33(4), pp. 807-822. https://doi.org/10.1007/s11390-018-185

  1. The origin of Equations (1) and (3) needs to be supported by a reference, which must be added.

We cited, in both cases, the reference [18].

  1. I think itís best to put both algorithms into explicit steps that enable the reader to visualize the code. Is this accessible to the authors?

We explained the application of formula (2) in three steps: i) Computation of the ranges of A and B; ii) Intersection between A and B; and iii) Computation of  (Abreviatly , aff). We also included the algorithm concerning the computation of the local generalized affinity coefficient,  between a pair of statistical units, ? and ?′ (?, ? ′ = 1(1)N), concerning the variable Yj , according to formula (2).  We also clarified the explanation of the Algorithm 2.

  1. Tables 1 and 2 need further clarification.

In the third row after Table 1, we added the following information:

…Thus, each entry of this table is defined as a closed and bounded interval and are often used to represent a quantity which may vary between an upper boundary and a lower boundary.

Before Table 2, we clarified the text in the following way:

… where  symbolizes the interval range (difference between the upper and the lower boundaries of the corresponding interval).

In a second step, ….in terms of the ranges of the intersection between the initial interval,  (k=1(1)N) and each of the mj elementary subintervals used in the decomposition of the domain of the variable Yj (j=1(1)p)). Thus, the vector [[a1j, b1j], , [akj, bkj], , [aNj, bNj]]T, corresponding to Yj in Table 1, is replaced by an appropriate subtable, as shown in Table 2.

  1. In addition to the fantastic conclusion and discussion section, I think it would be better to include the most notable results as bright spots at the end of the section.

Done.

Reviewer 2 Report

The work is very interesting and it is belong a new methodology as Symbolic Data Analysis and consider interesting new approaches in the field. This study conducts a comparative examination of two algorithms used in the computation of the weighted generalized affinity coefficient for interval data. The methods under investigation utilize hierarchical cluster analysis and probabilistic aggregation criteria.
Along the author's presentation and discussion, the algorithms exhibited the capability to identify persistent clusters within macro-level data and demonstrated proficiency in managing extensive microdata repositories after their conversion into macro-level data. In some scenarios, one algorithm exhibits superior performance compared to the other. 

The work is innovative and grounded on previous scientific results. The theme is very innovative and merit attention for the relevant implications on big data analysis and on the same development of the techniques in this literature.

Yet, the results are not fully convincing for the fact:

- the visual presentation of the clustering (the dendrograms in particular) can be improved. It is a relevant fact that software development is a relevant issue, but an effort should be done on presenting results on a best visual graphics way.

- the authors use some known dataset, it is better use a new dataset which practically consider a real case, with a real solution by the methodology used in a concrete real context

- A relevant mention is important on the software used and on the possible use of the algorithms considered in R, Python or other language.

The quality of the English can be improved.

Author Response

The work is very interesting and it is belong a new methodology as Symbolic Data Analysis and consider interesting new approaches in the field. This study conducts a comparative examination of two algorithms used in the computation of the weighted generalized affinity coefficient for interval data. The methods under investigation utilize hierarchical cluster analysis and probabilistic aggregation criteria.
Along the author's presentation and discussion, the algorithms exhibited the capability to identify persistent clusters within macro-level data and demonstrated proficiency in managing extensive microdata repositories after their conversion into macro-level data. In some scenarios, one algorithm exhibits superior performance compared to the other. 

The work is innovative and grounded on previous scientific results. The theme is very innovative and merit attention for the relevant implications on big data analysis and on the same development of the techniques in this literature.

Yet, the results are not fully convincing for the fact:

- the visual presentation of the clustering (the dendrograms in particular) can be improved. It is a relevant fact that software development is a relevant issue, but an effort should be done on presenting results on a best visual graphics way.

The software used is efficient, but the visual aspect of the dendrograms can be improved. However, in general, the dendrograms provided by this software are somewhat easier to interpret than those provided by, for example, IBM SPSS Statistics. We intend to develop an articulation between our program and the R software, for use in future work.

- the authors use some known dataset, it is better use a new dataset which practically consider a real case, with a real solution by the methodology used in a concrete real context

We have used some real data (e.g., reference [33] of the present paper). Currently, we are working on two new real databases, together with other authors, with a view to being used in other articles.  The main objective of this article is to compare the two algorithms. Therefore, we use two well-known data sets.

- A relevant mention is important on the software used and on the possible use of the algorithms considered in R, Python or other language.

We used the Cluster program, developed by members of our working group, but we are working in a possible articulation of Cluster software with the R language. 

Reviewer 3 Report

In the presented article, the authors propose comparison between two algorithms for computing the weighted generalized affinity coefficient in the case of interval data.

The problem solved in the manuscript is quite interesting, but the study needs to be refined and expanded.

Make correct numbering of sections (two second sections).

In section 2, add a description of ways and methods of solving the described problem. Specify the disadvantages of the existing and previously developed solution methods.

Taking into account the topic of cluster analysis and the speed of appearance of new methods and algorithms, the References contain the most "new" articles from 2015-2016. It is necessary to show more recent studies.

Source 4 is not referenced in the text. There are references to several sources at the same time in the text, made not according to the rules (for example, line 100).

Algorithms 1 and 2, besides the description, it is desirable to show classically, step by step.

The issue of clustering quality (accuracy) was not considered at all? And the number of clusters is also set at the beginning, not determined.

How many times were the algorithms run to obtain the results and were the mean or median values taken at that time? What is the computation time or number of iterations for an acceptable convergence criterion for the algorithms?

The main criteria for the "success" of an algorithm are:

1) the time for which the algorithm solved the problem (the faster, the better);

2) the amount of memory required for the algorithm to work (the smaller, the better).

Describe in the text what exactly is better in the algorithms under consideration.

The second dataset for research is perhaps not quite right, as two of the 4 clusters have only one element each.

For a full-fledged conclusion of the comparison of the two algorithms it is necessary to cite the results of other methods as well, showing that the two considered are among the best or even the only ones of their kind for such studies. And add at least one more dataset, taking into account the mentioned peculiarity of the second dataset.

What is missing is a summary table of results of all algorithms for comparison. And the comparison itself should be shown in figures (e.g., so many per cent more accurate, so much faster time).

I would also like to know the parameters of the equipment on which the calculations were performed.

Line 314 - Figure 2, instead of Figure 1.

Author Response

In the presented article, the authors propose comparison between two algorithms for computing the weighted generalized affinity coefficient in the case of interval data.

The problem solved in the manuscript is quite interesting, but the study needs to be refined and expanded.

Make correct numbering of sections (two second sections).

Done

In section 2, add a description of ways and methods of solving the described problem. Specify the disadvantages of the existing and previously developed solution methods.

We added the following paragraph:

Several clustering algorithms have been proposed in the literature. Ezugwu et al. ([12]) present a systematic review of traditional and state-of-the-art clustering techniques for different domains. Moreover, some dissimilarity measures for interval data can be found in the literature concerning the symbolic data (e.g., [8], [13], [14-17]), as well as some similarity measures which can deal with the case of interval data (e.g., [18-21], Ramos-Guajardo, 2022).

Taking into account the topic of cluster analysis and the speed of appearance of new methods and algorithms, the References contain the most "new" articles from 2015-2016. It is necessary to show more recent studies.

We added four references, namely [8, 9, 12, 13]:

Goswami, J.P.; Mahanta, A.K. Interval Data Clustering. IOSR Journal of Computer Engineering (IOSR-JCE) 2020, 22(6), Ser. 1, pp. 45-50. doi: 10.9790/0661-2206014550

Ramos-Guajardo, A.B. A hierarchical clustering method for random intervals based on a similarity measure. Comput Stat 2022, 37, 229-261.

Ezugwu, A.E.; Ikotun, A.M.; Oyelade, O.O..; Abualigah, L.; Agushaka , J.O.; Eke, C.I., Akinyelu, A. S. A comprehensive survey of clustering algorithms: State-of-the-art machine learning applications, taxonomy, challenges, and future research prospects. Engineering Applications of Artificial Intelligence 2022, 110, 104743. https://doi.org/10.1016/j.engappai.2022.10474

Xu, X.; Lu, J..; Wang, W. Hierarchical clustering of complex symbolic data and application for emitter. J. Comput. Sci. TechnoL. 2018, 33(4), pp. 807-822. https://doi.org/10.1007/s11390-018-185

Source 4 is not referenced in the text. There are references to several sources at the same time in the text, made not according to the rules (for example, line 100).

Now Source 4 is referenced in the text.

Algorithms 1 and 2, besides the description, it is desirable to show classically, step by step.

 We explained the application of formula (2) in three steps: i) Computation of the ranges of A and B; ii) Intersection between A and B; and iii) Computation of  (Abreviatly , aff). We also included the algorithm concerning the computation of the local generalized affinity coefficient,  between a pair of statistical units, ? and ?′ (?, ? ′ = 1(1)N), concerning the variable Yj , according to formula (2).  We also clarified the explanation of the Algorithm 2.

The issue of clustering quality (accuracy) was not considered at all? And the number of clusters is also set at the beginning, not determined

This is a very important topic. However, taking in consideration the main objective of this article (comparison of two algorithms), we chose to use two well-known data sets in terms of clustering structure.

How many times were the algorithms run to obtain the results and were the mean or median values taken at that time? What is the computation time or number of iterations for an acceptable convergence criterion for the algorithms?

The main criteria for the "success" of an algorithm are:

1) the time for which the algorithm solved the problem (the faster, the better);

2) the amount of memory required for the algorithm to work (the smaller, the better).

Describe in the text what exactly is better in the algorithms under consideration.

These aspects are very interesting and could be the subject of a new article (the authors thank you for the suggestion). However, these aspects go beyond the objectives of this paper.

The second dataset for research is perhaps not quite right, as two of the 4 clusters have only one element each.

The intuitive partition, carried out by a group of human observers, resulted in four clusters of cities (e.g., ([40])): {2, 3, 4, 5, 6, 8, 11, 12, 15, 17, 19, 22, 23, 29, 31}; {0, 1, 7, 9, 10, 13, 14, 16, 20, 21, 24, 25, 26, 27, 28, 30, 33, 34, 35, 36}; {18}; {32}. In fact, two of the 4 clusters have only one element each.

For a full-fledged conclusion of the comparison of the two algorithms it is necessary to cite the results of other methods as well, showing that the two considered are among the best or even the only ones of their kind for such studies. And add at least one more dataset, taking into account the mentioned peculiarity of the second dataset.

We use two well-known data sets in terms of clustering structure.

What is missing is a summary table of results of all algorithms for comparison. And the comparison itself should be shown in figures (e.g., so many per cent more accurate, so much faster time).

I would also like to know the parameters of the equipment on which the calculations were performed.

Our main objective is to compare the 2 algorithms in terms of the obtained clustering structures. The other aspects indicated are very interesting concerning the guidelines for future work.

.Line 314 - Figure 2, instead of Figure 1.

Done

Round 2

Reviewer 2 Report

Main my points presented are post-poned by the authors to future papers. It is reasonable in absence of relevant algorithms to be used in practice, which can be a work by itself (in fact exists a relevant literature in last years in Statistics names "Statistical Software" studies on the creation and development of programs\packages\libreries to perform efficiently statistical analysis based on existing methods or innovative one). The "real case" used it is reasonable and can be considered as an evidence to "real use" of the methods in "real" problems. The research, at this actual stage of development is now fine. A revision of the English is yet necessary to improve readability.

A revision of the English is reasonable.

Author Response

This new revised version of  our  manuscript “has undergone English language editing by MDPI. The text has been checked for correct use of grammar and common technical terms, and edited to a level suitable for reporting research in a scholarly journal”.  

Reviewer 3 Report

The authors have amended and clarified in line with my suggestions.

Some of my comments are considered by the authors as future studies and articles. At this stage of comparing the two algorithms, the research is sufficient.

The manuscript requires minor revision and correction of minor errors, e.g. in source 12 the DOI should end with 104743.

Author Response

This new revised version of  our manuscript “has undergone English language editing by MDPI. The text has been checked for correct use of grammar and common technical terms, and edited to a level suitable for reporting research in a scholarly journal”.  

We corrected the doi concerning Reference [12].

Round 3

Reviewer 2 Report

The work is now completed. Well done.

Reviewer 3 Report

The authors have made changes and clarifications in accordance with my comments.